# Oxygen-catalysed sequential singlet fission

Nikolaus Wollscheid [1,2,4], J. Luis Pérez Lustres[1,2,4], Oskar Kefer[1,2], Sebastian Hahn[2,3], Victor Brosius[2,3], Uwe H.F. Bunz[2,3], Marcus Motzkus[1,2]* & Tiago Buckup [1,2]*

Singlet fission is the photoinduced conversion of a singlet exciton into two triplet states of half-energy. This multiplication mechanism has been successfully applied to improve the efficiency of single-junction solar cells in the visible spectral range. Here we show that singlet fission may also occur via a sequential mechanism, where the two triplet states are generated consecutively by exploiting oxygen as a catalyst. This sequential formation of carriers is demonstrated for two acene-like molecules in solution. First, energy transfer from the excited acene to triplet oxygen yields one triplet acene and singlet oxygen. In the second stage, singlet oxygen combines with a ground-state acene to complete singlet fission. This yields a second triplet molecule. The sequential mechanism accounts for approximately 40% of the triplet quantum yield in the studied molecules; this process occurs in dilute solutions and under atmospheric conditions, where the single-step SF mechanism is inactive.

[1] Physikalisch Chemisches Institut, Ruprecht-Karls University, Im Neuenheimer Feld 229, 69120 Heidelberg, Germany. [2] Centre for Advanced Materials, University of Heidelberg, Im Neuenheimer Feld 225, 69120 Heidelberg, Germany. [3] Organisch Chemisches Institut, Ruprecht-Karls University, Im Neuenheimer Feld 270, 69120 Heidelberg, Germany. [4] These authors contributed equally: Nikolaus Wollscheid, J. Luis Pérez Lustres. *email: marcus.motzkus@pci.uni-heidelberg.de; tiago.buckup@pci.uni-heidelberg.de

Cost-effective avenues for solar energy conversion are in high demand[1–3]. Architectures based on organic materials and amorphous semiconductors are interesting due to an increased charge/photon ratio, efficient use of the solar spectrum and low fabrication costs[4–6]. Central to this advance has been the way in which high-energy photons ($E_{photon} \gg E_{bandgap}$) are brought into play. According to the Shockley–Queisser limit[7], the maximum efficiency of a single-junction solar cell is ~34%, basically because the photon energy exceeding the bandgap is lost as heat. In turn, multiplication mechanisms[8,9] invest high-energy photons to generate two low-energy carriers, leading to external quantum efficiencies exceeding 100%[10,11]. Therefore, maximising charge extraction from the low-energy carriers while minimising parallel loss channels is one of the key issues that must be resolved to boost conversion efficiency in solar cells.

In carrier multiplication by singlet fission (SF), the first excited singlet-state ($S_1$) forms two triplet states ($T_1$) upon close inter-action between a $S_1$-excited state and a ground-state ($S_0$) molecule of the same kind (Fig. 1a). The ensuing[1]($T_1T_1$) state, the correlated triplet pair, preserves the global singlet character, and the process, contrary to intersystem crossing, is spin allowed. SF may occur on ultrafast timescales if important conditions are fulfilled. First, energy conservation demands that the triplet states have approximately half the energy of the excited singlet-state, $E(S_1)$[8,12,13]. Second, the reacting $S_0 – S_1$ molecules have to be at contact distances. In films, SF was found to occur at timescales ranging from fs to ns, depending on the energy detuning $E(S_1) - 2 \cdot E(T_1)$ and the strength of the coupling matrix element[14]. This has also been observed in solution, where 6,13-bis(triisopropylsilylethenyl)-pentacene (TIPS-Pn, Fig. 1c) undergoes SF in the diffusion-control limit[15]. Therefore, in favourable cases, SF outcompetes singlet-state deactivation by fluorescence and internal conversion, with intersystem crossing being of no concern for such a large $E(S_1) - E(T_1)$ energy bias: photon-to-electron conversion efficiencies close to 200% are achievable in the visible spectral range if charges can be efficiently extracted from the long-living triplet states.

This work uses transient absorption and time-resolved photoluminescence measurements to demonstrate a sequential SF mechanism with ground-state triplet molecular oxygen $^3O_2$ acting as a catalyst (Fig. 1b). In this process, two triplet states are produced in a step-by-step manner in solution. The sequential mechanism accounts for ~40% of the triplet quantum yield in TIPS-pentacene, as well as tetrachlorphenazinothiadiazole, and is transferrable to all SF molecules with suitable $S_1$ and $T_1$ energies.

## Results

**Overview of the catalysed singlet fission.** In the catalysed sequential singlet fission reaction (Fig. 1b), two steps are distinguished. In an initial energy transfer stage, the chromophore is excited to the first excited singlet-state $S_1$ by an incoming photon. Subsequently, it crosses to its $T_1$ state in a spin-allowed singlet-triplet annihilation involving the excited chromophore and $^3O_2$. The latter is thereby excited to its lowest excited singlet-state ($^1O_2$). This process is a well-established method used in the photosensitised production of $^1O_2$ and is exothermic whenever the singlet-triplet energy gap of the chromophore exceeds the singlet-triplet energy gap of $O_2$ (0.977 eV)[16]:

$$E(S_1) - E(T_1) \geq 0.977 \, \text{eV} \qquad (1)$$

This reaction occurs in the diffusion-control limit for acenes[16,17]. Additionally, the energy transfer to the long-lived $^1O_2$ species (decay time of 26.7 μs in benzene)[18] greatly increases the diffusion length of the excited singlet species from 3.6 nm to 246 nm (see Supplementary Note 1). In the second stage, a heterogeneous

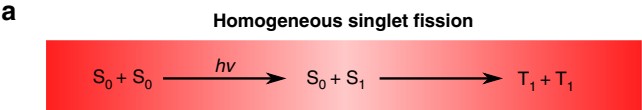

**a** Homogeneous singlet fission

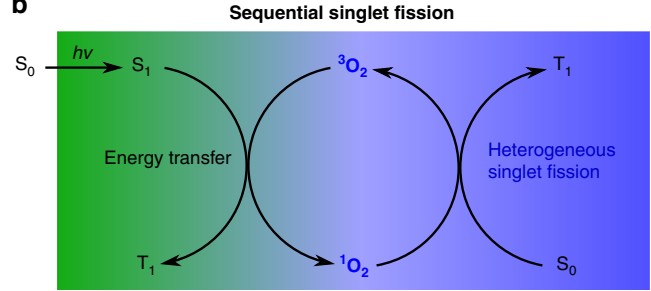

**b** Sequential singlet fission

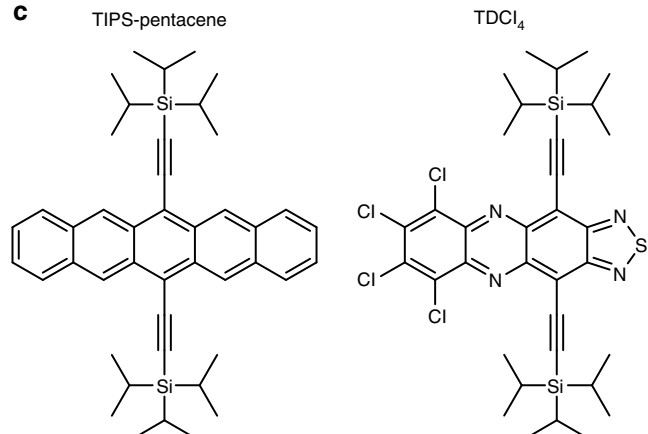

**c**  TIPS-pentacene          TDCl$_4$

**Fig. 1** Overview of reaction mechanisms and molecular structures. **a** Reaction scheme for homogeneous singlet fission. Upon excitation, a singlet exciton interacts with a ground-state chromophore to yield two triplet states. **b** Photocycle for oxygen-catalysed sequential singlet fission. The excited singlet chromophore transfers part of its energy to $^3O_2$ in a spin-allowed process, which yields the reactive $^1O_2$ species and the triplet state of the SF chromophore. Next, an additional ground-state chromophore is sensitised by $^1O_2$ in a heterogeneous singlet fission process, regenerating $^3O_2$. On the whole, sequential SF forms two triplet states from one absorbed photon, just as in homogeneous SF. **c** Chemical structures of TIPS-pentacene and TDCl$_4$. Their first absorption band is located between 600 and 700 nm, and their most stable triplet state is located at approximately half the $S_0S_1$ bandgap, which is nearly isoenergetic with $^1O_2$

SF process occurs, i.e., the reaction of two singlet states of different kinds to yield their respective triplets: $^1O_2$ interacts with a chromophore in the singlet ground state, resulting in the $T_1$ chromophore and the $^3O_2$ ground state of oxygen. The later stage will also occur in the diffusion-control limit if the chromophore triplet level lies below the energy of singlet oxygen:

$$0.977 \, \text{eV} \geq E(T_1) \qquad (2)$$

As a fortunate consequence, triplet quenching by $^3O_2$ becomes endothermic. Furthermore, both steps are efficient because $^3O_2$ and $S_0$ are in excess (mM concentrations). Adding 1 and 2 yields the well-known energy requirement for SF:

$$E(S_1) - 2 \, E(T_1) \geq 0 \qquad (3)$$

Thus, any chromophore that shows oxygen-assisted sequential SF also undergoes homogeneous SF. However, the correlated triplet

pair discussed for solid-state SF[19–22] does not need to be considered in dilute solution, as $^1(T_1{}^3O_2)$ does not accumulate in the diffusion-controlled process. In the following, the interplay of homogeneous and sequential SF and the effects on the overall dynamics, particularly the rate constants, are discussed.

After excitation, homogeneous SF and energy transfer to $^3O_2$ (with bimolecular rate constants $k_{SF}$ and $k_1$, respectively) occur simultaneously out of the same $S_1$ species. For both reactions, the respective reaction partner for the $S_1$ species ($S_0$ and $^3O_2$) is in great excess in comparison with the small concentration of the photoexcited $S_1$ state. Thus, their concentrations can be assumed to remain constant over time, resulting in a pseudo-first-order reaction for both processes. As both SF processes compete with each other as well as unimolecular singlet relaxation ($k_R$), their respective rate constants are observed collectively within the total $S_1$ decay rate ($k_{tot}$). Thus, the total $S_1$ decay rate constant can be written as

$$k_{tot} = k_R + k_1 [{}^3O_2] + k_{SF}[S_0] \qquad (4)$$

It is possible to disentangle under excess concentrations the individual values by the bimolecular nature of homogeneous SF (Fig. 2b, red arrow) and energy transfer between the chromophore and the molecular oxygen ($^3O_2$) (Fig. 2a, green arrow). The rates for both processes depend on the concentration of the excited chromophore, as well as of their respective reaction partner: the ground-state chromophore ($S_0$) and molecular oxygen ($^3O_2$), respectively, which are experimentally known. In contrast, the unimolecular singlet decay rate constant ($k_R$) does not depend on the concentrations of the ground-state chromophore or molecular oxygen and explains the intercept of $k_{tot}$ as a function of the concentrations of both reaction partners (chromophore $S_0$ and triplet oxygen).

The homogeneous SF rate constant is obtained as the slope of the total singlet decay rate with varying chromophore concentration at constant oxygen concentration. Similarly, $k_1$ can be determined by keeping $[S_0]$ constant and varying $[{}^3O_2]$. The individual timescales for these two sub-reactions can be vastly different, depending on the respective concentrations of the reaction partners (see Supplementary Note 6). The heterogeneous SF reaction of $^1O_2$ with a ground-state chromophore depends on the concentration of the latter and thus occurs with a rate of $k_2 [S_0]$. Owing to the long lifetime of $^1O_2$ (26.7 μs in benzene)[18], a quantitative reaction is expected.

The sequential mechanism (Fig. 1b) is demonstrated here for two different molecules: tetrachloro-phenazinothiadiazole (TDCl$_4$)[14,23] and TIPS-Pn (Fig. 1c). The latter has been paradigmatic for homogeneous SF. The first excited singlet-state of TIPS-Pn is located at 2.02 eV, and the first triplet state is 0.85 eV above its ground state (Fig. 2)[14]. Thus, the energy transfer step sets 0.19 eV free. Subsequently, the heterogeneous SF stage has a driving force of 0.13 eV, resulting in an overall exothermicity of 0.32 eV. As described above, the triplet pair can also be formed via homogeneous SF. The combination of both SF mechanisms (Fig. 2) leads to a characteristic dynamics of the involved species, which can be first simulated with literature values (Supplementary Table 1 and 3)[15,17,24,25].

The dependence of these rate constants as well as of the quantum yield on the chromophore concentration is depicted in Fig. 2c within a kinetic model that takes into account sequential and homogeneous SF. In this regard, three distinct regions can be identified: low, intermediate and high TIPS-Pn concentrations. At low concentrations, homogeneous SF is negligible, resulting in a high quantum yield for competing energy transfer and heterogeneous SF. Furthermore, the low $S_0$ concentration allows for the isolation of the heterogeneous SF rate constant. At high TIPS-Pn concentrations, $k_{SF}$ outcompetes the other singlet decay

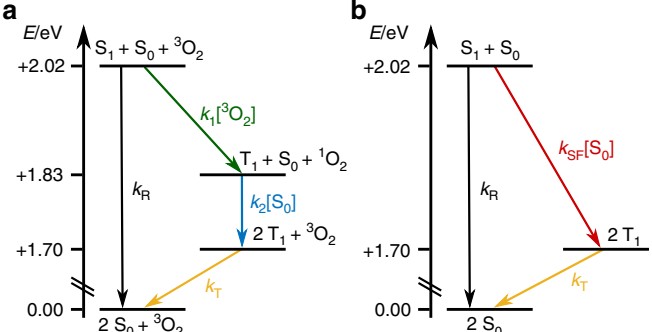

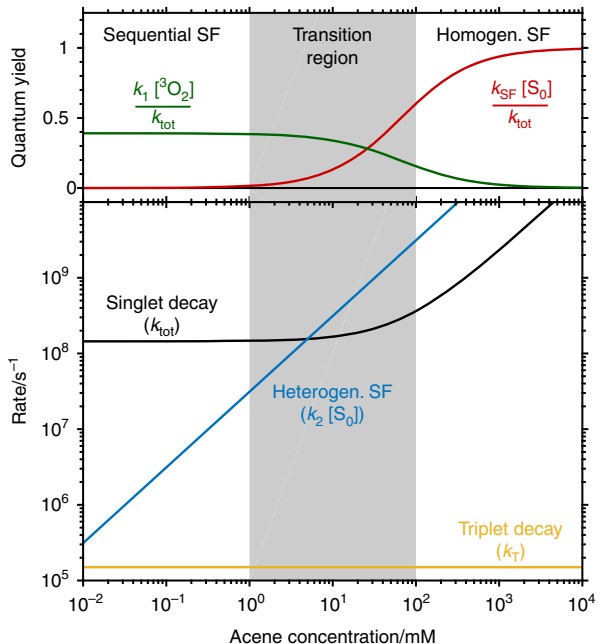

**Fig. 2** Reaction mechanisms and expected concentration dependence. **a** Scheme for the mechanism of oxygen-catalysed singlet fission of TIPS-Pn in solution with rate constants. An identical mechanism is proposed for TDCl$_4$. Owing to the relative energies of $S_1$ and $T_1$ compared to $^1O_2$, the individual steps of sequential singlet fission are exothermic. **b** Scheme for the mechanism of homogeneous singlet fission in TIPS-Pn with corresponding rate constants. **c** Upper panel: expected chromophore concentration dependence for the quantum yields of homogeneous (red) and sequential (green) SF. Lower panel: expected chromophore concentration dependence for rate constants of the individual processes. The association of an excited and ground-state chromophores ($k_{SF}$) is the rate-limiting step of homogeneous SF. SF is outweighed by concentration independent, non-SF relaxation pathways (i.e., fluorescence, internal conversion and, to a minor extent, intersystem crossing) for low chromophore concentrations and in the absence of oxygen (black, $\mathbf{k_{tot} \approx k_R}$). At higher chromophore concentrations, $k_{tot} \approx k_{SF}[S_0]$ (black). In contrast, the rate of the first step in sequential SF, that is, the energy transfer ($k_1[{}^3O_2]$), is not affected by the chromophore concentration because $^3O_2$ is in excess compared to the excited chromophore. However, the subsequent heterogeneous singlet fission (blue, $\mathbf{k_2[S_0]}$) exhibits a pseudo-first-order reaction rate and thus a linear dependence on the chromophore ground-state concentration. Finally, regardless of how the triplet is formed, its relaxation rate remains unaffected by the chromophore concentration (yellow, $\mathbf{k_T}$). Source Data are provided as a Source Data file

mechanisms, resulting in $k_{tot} \approx k_{SF} [S_0]$. At intermediate concentrations, both SF processes have similar rate constants and thus occur in parallel. The decay of the produced triplet, $k_T$, is concentration independent and is observable in all circumstances.

**Application to TIPS-pentacene.** The broadband transient absorption (TA) signal of TIPS-Pn is analysed in the concentration range from 0.02 to 160 mM to expose the transition from the sequential to the homogeneous SF mechanism as the concentration rises. Essentially, TA monitors the change in the absorption seen by a broadband probe beam as induced by an optical excitation pulse interacting with the sample at some given earlier time. The TA measurements cover the UV–Vis spectral window completely and address timescales from ns to μs. Whereas the TA experiment monitors the time evolution of all reaction partners simultaneously, time-correlated single-photon counting fluorescence measurements track the time evolution of the first excited singlet-state of the chromophore, which is the only fluorescent species (Supplementary Fig. 4). The combined analysis of both data sets provides solid ground to examine the mechanism proposed here. TDCl$_4$, which is structurally different from TIPS-Pn but otherwise fulfils the energetic requirements for SF and oxygen-catalysed SF, should provide compelling evidence[14].

The TA signal of TIPS-Pn for a 0.5 mM concentration is shown in Fig. 3a. The TA spectra show the positive contribution of excited-state absorption (ESA), which can be recognised in the range between 350 and 680 nm, and the negative contributions of bleach and stimulated emission (SE). The bleach arises from molecules being promoted to excited states, which consequently depopulates the ground state. As TA monitors absorption differences, a decrease in the signal is observed, with its spectral contributions being equal to the UV–Vis absorption spectrum of the chromophore with a negative sign. In turn, SE is the absorption decrease as a consequence of the fluorescence emission stimulated by the incoming probe beam. Thus, SE shows a negative sign, and its shape nearly mirrors the stationary fluorescence emission spectrum[26]. The negative contributions of bleach and SE are observed between 680 and 775 nm. At longer wavelengths, weak ESA overcompensates the red tail of SE, and the signal becomes positive again. At any probe wavelength, the signal amplitude is proportional to the excited-state concentration, which makes TA an excellent method for tracing photoinduced dynamics.

The TA signal of TIPS-Pn evolves on the ns time scale for a concentration of 0.5 mM. Spectral shape and amplitude remain constant in the fs-ps time window. Hence, according to previous reports[27], the earliest TA spectrum is assigned to the S$_1$ state. It shows a prominent ESA peak at 445 nm together with weaker features at 507, 533 and 572 nm. Bleach and SE explain the negative peaks at 593, 644 and 702 nm. This TA spectrum decays on the same time scale as that of fluorescence, giving rise to a long-lived component (green spectrum in Fig. 3a) with ESA peaks at 465 and 498 nm, bleach at 590 and 642 nm and no signature of SE. This last spectrum is assigned to the triplet state T$_1$ of TIPS-Pn[28]. There is no indication of additional states.

The time evolution of the TA signal at the μs timescale measured at the ESA maximum of the triplet state, 495 nm, is now addressed (Fig. 3b). Under atmospheric conditions, the signal rises with the leading edge of the instrument response function (IRF) and then decays with a ns time constant, which matches the fluorescence decay time. Intriguingly, a slow rising component is observed in the sub-μs range, which implies that the concentration of the triplet state continues to rise after the singlet-state has decayed completely. The signal reaches a maximum at ~0.4 μs and then decays slowly. Therefore, the signals were globally fitted to three exponential functions, and the associated amplitudes were extracted (see for example Supplementary Figs. 9–14). The results for atmospheric conditions and degassed solutions are compared. Two experimental facts are most remarkable: Firstly, the T$_1$ signal increases until it reaches its

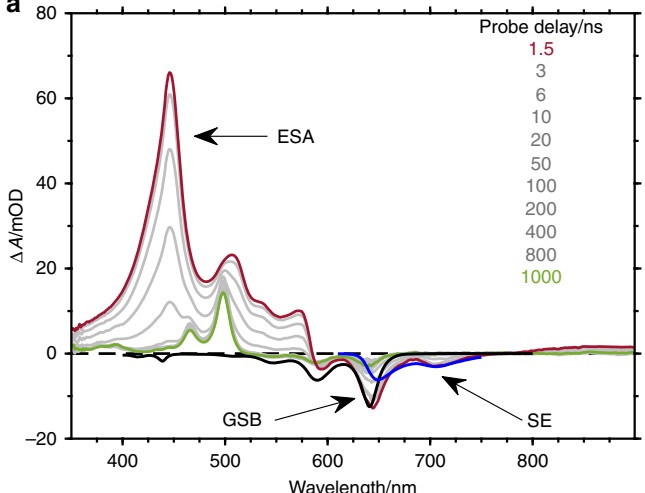

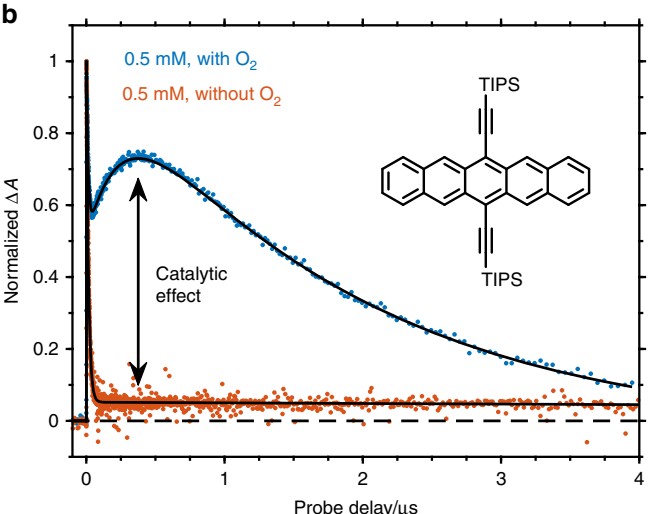

**Fig. 3** Spectral evolution in time and oxygen dependence of kinetic traces. **a** Spectral evolution of a 0.5 mM solution of TIPS-Pn in THF. In the initial 6 ns, the transient spectrum is determined by S$_1$ excited-state absorption with a maximum at 450 nm. Additionally, signatures of ground-state bleach and stimulated emission are observed at 640 and 700 nm, respectively, in good agreement with the absorption and emission spectra (black and blue lines). After 50 ns, a triplet ESA with a maximum at 500 nm can be observed in lieu of the singlet. No further spectral changes are observed. **b** Selected transients at 500 nm probe wavelength. The kinetic traces for a 0.5 mM solution of TIPS-Pn (see inset) show a stark contrast when using THF stored under atmospheric conditions (blue) compared to degassed THF (red). At the maximum of the triplet ESA (500 nm), the initial fast decay of the singlet is observed. In the case of degassed THF, the amplitude of the triplet is very weak at μs delays. However, in the solution containing oxygen, after the fast singlet decay, the triplet amplitude rises slowly until a local maximum is observed at a probe delay of ≈ 0.4 μs. This behaviour indicates an oxygen-catalysed SF process. Additionally, the triplet decays much faster than in the degassed solution ($\tau_{deg}$ = 28.2 μs; $\tau_{O_2}$ = 1.8 μs) due to further interaction with oxygen (see Supplementary Note 5 for more details). Source Data are provided as a Source Data file

maximum at ~0.4 μs under ambient conditions. This implies an intermediate in the conversion from S$_1$ to T$_1$. However, this intermediate is not obvious in the TA spectra, as no spectral component other than the singlet and the triplet state change of TIPS-Pn is observed. Additionally, the maximal triplet concentration decreases to ~7% in the absence of oxygen but under

otherwise equal experimental conditions (Fig. 3b and Supplementary Fig. 5). At the same time, the fluorescence lifetime increases from 12.3 to 14.9 ns upon degassing the solution (Supplementary Fig. 4), indicating that oxygen quenches the $S_1$ state. Both findings together indicate that this two-stage build-up of the $T_1$ state can neither be explained by homogeneous SF nor intersystem crossing, for which no dependence on oxygen concentration can be anticipated.

Additional TA measurements were carried out under ambient conditions for chromophore concentrations in the range from 0.02 to 160 mM. These measurements provide a global view of the singlet deactivation processes, leading to the population of the triplet manifold of TIPS-Pn. To quantify the results, a global analysis was carried out, shown exemplary at the maximum of the $T_1$ ESA (500 nm, Fig. 4). Here, the dynamics depend on the $S_0$ concentration. Within the range of $0.1\,mM \leq [S_0] \leq 6\,mM$, the TA signal evolves three-exponentially in time. A sharp initial intensity drop within $< 0.1\,\mu s$ is observed, corresponding to the decay of the $S_1$ state. Subsequently, a rising component in the $T_1$ signal as well as a local µs maximum can be identified. Increasing the chromophore concentration accelerates this component and thus shifts this local µs maximum to earlier probe delays. For concentrations above 10 mM, biexponential behaviour is recorded. First, the $S_1$ state decays single-exponentially, and the rate depends linearly on the TIPS-Pn concentration, as monitored by TA and time-resolved fluorescence (Supplementary Fig. 4). Second, the $T_1$ state rises as the $S_1$ state decays. The additional, slower, triplet rising component accelerates with the TIPS-Pn concentration. Finally, its relative contribution to the triplet dynamics grows with concentrations up to $\approx 1\,mM$ and then declines upon further concentration increase. Finally, the $T_1$ state always decays at the same rate, 1.5 µs on average.

## Discussion

The trend described above is shown in Fig. 5, from which all relevant rate constants are obtained by linear fitting of the experimental rate constants as a function of the chromophore concentration (Supplementary Table 3). The correspondence with the prediction outlined in Fig. 2 is noticeable. The so-obtained rate constants were used in the time-dependent concentrations deduced for the competing sequential and homogeneous SF mechanisms in Fig. 1 (see Supplementary Note 1). This information is then used in a global analysis[29,30] to transform by matrix algebra the spectral amplitudes of the fitted exponential functions, the decay-associated difference spectra (DADS), into the transient spectra of the $S_1$ and $T_1$ states. The latter are the species-associated-difference spectra (SADS), which are shown to be consistently the same for all concentrations (Supplementary Fig. 16). To summarise, the global fit confirms that the rate constants follow the concentration dependence predicted by the mechanisms in Fig. 2a.

The upper panel in Fig. 5 shows the quantum yield for energy transfer and homogeneous SF. Both processes ultimately lead to the formation of two triplets (see above). The upper panel reveals that homogeneous SF can reach quantum yields of 100% only at high acene concentrations on the order of 0.1–1 M. In contrast, sequential SF occurs even at very low concentrations, albeit with lower quantum yield. By increasing the concentration of $^3O_2$ from 1.81 mM (ambient conditions[25]) to 5.0 mM in a 1.5 mM TIPS-Pn solution, the quantum yield rises from 21.6% to 43.4% (see Supplementary Note 5). Apart from this strategy, the quantum yield could also be improved by finding other suitable catalysts and/or introducing chemical modifications to the SF chromophore, which favour covalent or non-covalent bonding between acene and catalyst.

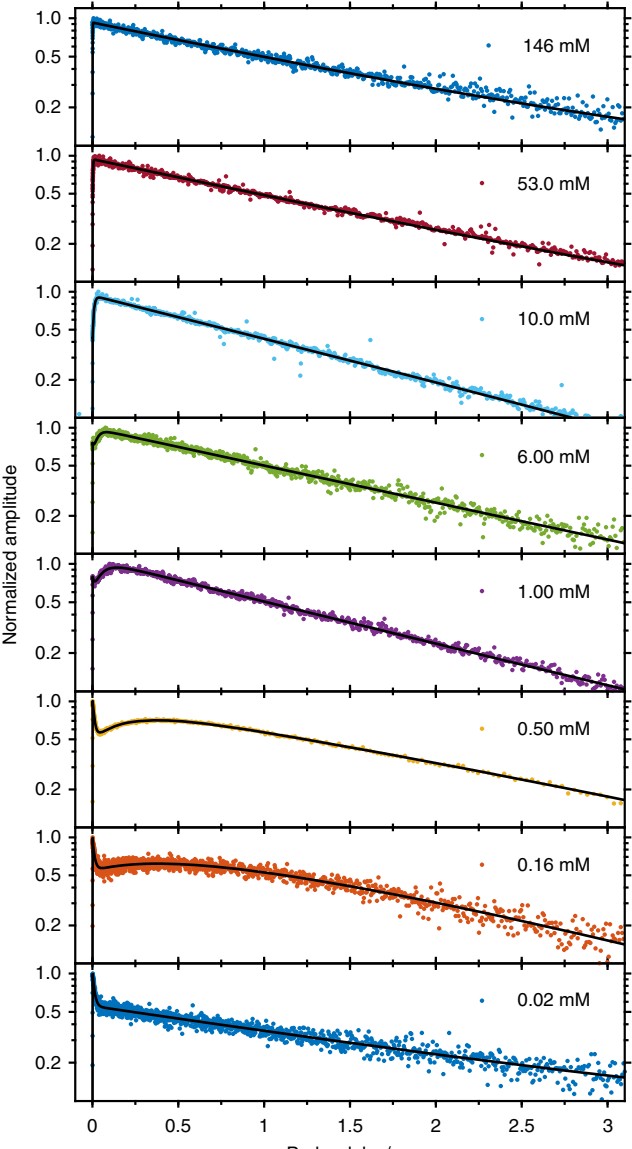

**Fig. 4** Kinetic traces of triplet ESA at selected concentrations with respective fit traces. The kinetic traces at a probing wavelength of 500 nm, that is, the maximum of the triplet ESA, are sensitive to the chromophore concentration. No µs local maximum is visible for the lowest concentration (0.02 mM), as sequential SF and triplet decay occur at comparable rates. At higher concentrations, the $S_0$ sensitisation process is accelerated, which explains a local maximum on the µs timescale and justifies the shift of that maximum towards shorter probe delays as the chromophore concentration is increased. For concentrations $\geq 10\,mM$, no maximum is visible as (i) the $^1O_2$ sensitisation is the rate-limiting step and (ii) the rate of homogeneous SF surpasses the rate of sequential SF. Source Data are provided as a Source Data file

Finally, the global analysis reports the SADS of the $S_1$ and $T_1$ states with the correct spectral shapes and relative amplitudes, from which the time-dependent concentrations of the two electronic states can be calculated accurately. Importantly, the method provides the SF quantum yields, for which no spectral decompositions or coarse assumptions about the absorption coefficients of the excited states have to be made. The analysis of the $TDCl_4$ TA measurements in the same concentration range provides further corroborating experimental evidence (Supplementary Fig. 15 and Supplementary Tables 2 and 4).

In conclusion, an oxygen-catalysed sequential SF process is demonstrated. This two-step mechanism is predicted to apply to a number of molecules for which SF was already demonstrated[14,19,20]. A singlet-triplet energy gap exceeding the

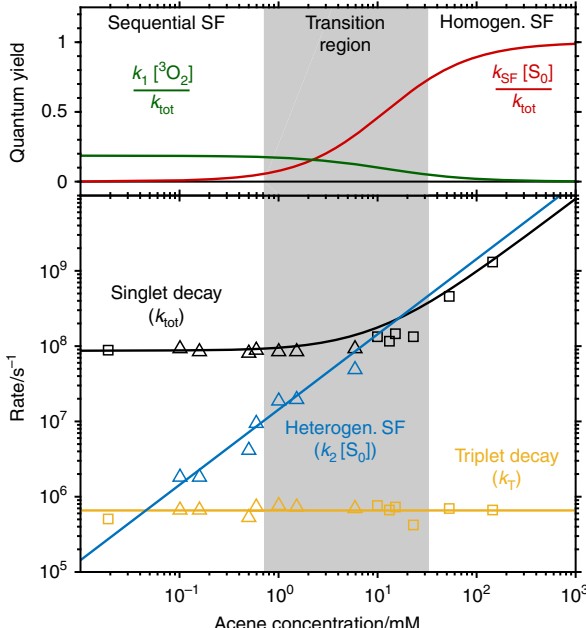

**Fig. 5** Singlet fission (SF) quantum yields and rate constants obtained by a global fit. The upper panel shows the acene concentration dependence of the SF quantum yield. The lower panel analyses the concentration dependence of the three rate constants explaining the time evolution of the singlet and triplet concentration. The experimental results perfectly reproduce the trend expected for the model of catalysed SF. For $c \leq 1$ mM, $k_R$ has a concentration-independent value of $(8.57 \pm 0.19) \times 10^7$/s, whereas for $c \geq 10$ mM, a linear fit of $k_{tot}$ (black) yields a slope of $(8.98 \pm 0.36) \times 10^9$/Ms, which is attributed to $k_{SF}$. As expected from the model (Fig. 4), an intermediate component— attributed to heterogeneous singlet fission—can be observed only in a concentration range of 0.1 mM < $c$ < 10 mM. The slope of the linear fit provides a value of $k_2 = (1.62 \pm 0.19) \times 10^{10}$/Ms. This value is in very good agreement with reported oxygen quenching rates[31]. The triplet decays with a concentration-independent rate of $k_T = (6.55 \pm 0.30) \times 10^5$/s. Source Data are provided as a Source Data file

relative energy of singlet oxygen is the main condition to fulfil (Fig. 6). For the highest efficiency of the second stage, i.e., the heterogeneous SF (Fig. 1b), the triplet state of the chromophore should lay below singlet oxygen in energy. This condition is not fulfilled for TIPS-Tn, which does not show triplet formation by heterogeneous SF (Supplementary Fig. 7). The mechanism and procedures described here open an uncomplicated strategy to determine relative singlet-triplet spectral amplitudes from which SF quantum yields can be calculated. The mechanism can be used to sensitise triplet states in high yield in an oxygen-rich atmosphere, which makes spectroscopy of the elusive triplet states feasible. Furthermore, sequential SF depicts a straightforward way to screen possible SF candidates absorbing ~650 nm, approximately twice the energy of singlet oxygen, as any such molecule fulfilling the prerequisites for sequential SF (1 and 2) automatically fulfils the energy requirement for homogeneous SF. Thus, a low time-resolution measurement in dilute solution and ambient conditions is sufficient to confirm the ability to undergo homogeneous SF.

## Methods

**Transient absorption measurements.** Transient absorption was measured in a commercial set-up (EOS from Ultrafast Systems). Pump beams were derived from a Ti:Sa amplified femtosecond laser system (Coherent Astrella, 4 kHz, 90 fs pulse duration, 1.5 mJ/pulse) using a commercial optical parametric amplifier (TOPAS Prime, Light Conversion) and probed using a triggerable on-demand super-continuum laser. The instrument response function reaches a FWHM of 350 ps. TIPS-Pn and TDCl$_4$ were excited at 640 and 620 nm, respectively.

**Stationary fluorescence measurements.** Steady-state fluorescence measurements for TIPS-Pn were carried out in a JASCO FP-8200 using a fluorescence cuvette with a triangular base (Starna, Type 24-SB/Q). Steady-state fluorescence measurements for TDCl$_4$ were performed at right-angle geometry in a LifeSpec-II Fluorometer (Edinburgh Instruments). A 10 μm thick fused silica cell (Starna, Type 48/Q) was placed at 45° with regard to the incident excitation beam (635 nm) and the optical axis of the collection optics. Thus, "front-face" fluorescence is measured with maximum avoidance of inner filter effects. All fluorescence spectra were corrected for instrumental factors.

**Time-resolved fluorescence measurements.** Fluorescence lifetime measurements were carried out with a Lifespec-II time-correlated single-photon counting (TCSPC) fluorescence lifetime spectrometer (Edinburgh Instruments). The sample was excited at 635 nm by a diode laser EPL-635 (Edinburgh Instruments) with a pulse width of 100 ps at selectable repetition rates. The time-resolution is estimated to be 10 ps after deconvolution.

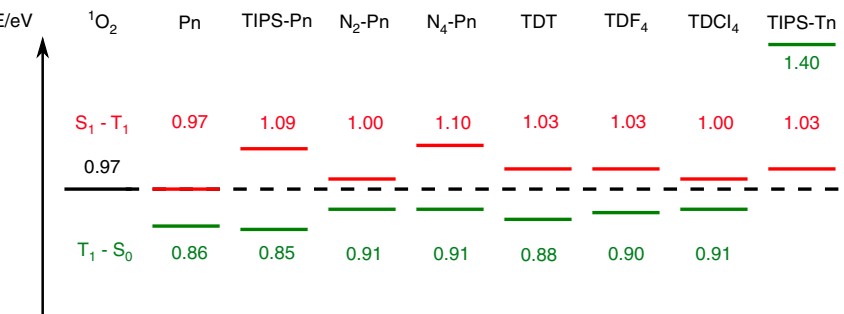

**Fig. 6** Relative energy differences for selected SF molecules. N$_2$-Pn, N$_4$-Pn, TDT, TDF$_4$, TDCl$_4$ and TIPS-Pn stand for diaza-TIPS-pentacene[19,20], tetraaza-TIPS-pentacene[19,20], phenazinothiadiazole[14], tetrafluorphenazinothiadiazole[14], tetrachlorphenazinothiadiazole[14] and TIPS-tetracene[14], respectively. The S$_1$–T$_1$ (red) and T$_1$–S$_0$ (green) energy gaps are shown and compared with the $^1O_2$–$^3O_2$ lowest energy gap of molecular oxygen (black). The feasibility of the two steps of catalysed SF, namely, $^3O_2$ and S$_0$ sensitisation, depends on the relative energies of the states involved. In the first step, the excited S$_1$ chromophore transfers part of its excitation energy to $^3O_2$. This reaction can occur only if $E(S_1 - T_1) \geq E(^1O_2 - {}^3O_2)$, i.e., if this energy difference is above the black dashed line. In the second step, a ground-state chromophore can be excited by energy transfer from $^1O_2$ if $E(T_1 - S_0) \leq E(^1O_2 - {}^3O_2)$, i.e., if this second energy gap is below the dashed black line. These prerequisites are fulfilled for many acenes exhibiting SF. According to this model, heterogeneous SF should not occur in TIPS-Tn, as the T$_1$ state lays higher than $^1O_2$ in energy (see Supplementary Fig. 7 for measurements done with and without O$_2$). Pentacene would in principle fulfil all conditions, but it is oxidised to an endoperoxide and thus not a suitable candidate[32]

**Sample preparation**. THF (Sigma-Aldrich) solutions were deaerated with five freeze-pump-thaw cycles in a home-built set-up. The pressure was below $5 \times 10^{-5}$ mbar after five cycles. At this point, solutions were vented with $N_2$. TIPS-Pn was purchased from Sigma-Aldrich and used without further purification. $TDCl_4$ was synthesised as described in the literature[33].

## Data availability

The source data underlying Figs. 2a–c, 3a, b, 4 and 5, and Supplementary Figs. 1–16 are provided as a Source Data file (https://heibox.uni-heidelberg.de/f/e2c38e9b070c42abb47c/).

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

## Acknowledgements

Financial support from the Deutsche Forschungsgemeinschaft (DFG) through the SFB 1249, Projects B04 and A01, is gratefully acknowledged.

## Author contributions

N.W., J.L.P.L., M.M. and T.B. were involved in the conception of the work. S.H. and U.B. synthesised TDCl4. N.W., J.L.P.L. and O.K. performed the measurements. V.B. synthesised TIPS-Tetracene as requested during the review process. N.W., J.L.P.L. and T.B. performed the literature and data analysis as well as discussed the results. N.W., J.L.P.L. and T.B. wrote the manuscript. All authors approved the submitted version of the manuscript.

## Competing interests

The authors declare no competing interests.
