## [Peer Review File · Nature Communications]

Reviewers' comments:

Reviewer #1 (Remarks to the Author):

The authors report very interesting observations of singlet fission in solution catalyzed by oxygen. In my opinion, the mechanism they propose for their explanation is convincing. I recommend that the results be published.

However, a thorough rewrite of the manuscript is required first. The present text is about twice longer than it needs to be and strikes me as confusing, repetitive, and generally disorganized. The English style will require attention, too. For instance, about half of the hyphens should disappear, along with phrases such as "Therefore, in favourable cases, SF outcompetes singlet state deactivation by fluorescence and internal conversion being intersystem-crossing of no concern for such a big $E(S) - E(T)$ energy bias ..."

Reviewer #2 (Remarks to the Author):

In the manuscript by N. Wollscheid et al. oxygen-mediated singlet fission in acene solutions is discussed. Energy transfer from a photoexcited acene to a triplet oxygen, followed by heterogeneous singlet fission between a singlet oxygen and a ground state acene yields two triplet excitons on two acenes.

The topic is of interest and the presented study differs significantly from other studies in the field and, thus, will contribute to the discussion of singlet fission pathways in organic semiconductors. While the data are in general well presented, there are some points which are unclear to me:

1) Commonly, the term singlet fission is used for the photophysical process in which two molecules are involved and interact directly (one in the ground state, the other in the excited state). Initially, both molecules are in their singlet state and in the (excited) triplet state after completion of the singlet fission process. Here, the two acenes do not directly interact, which justifies the term "oxygen-catalysed singlet fission". However, I am wondering, if the second step can still be considered as a heterogeneous singlet fission process, since in this case the oxygen, first, is not photoexcited and, second, returns to its (triplet) ground state after interaction with the ground state acene. In my view, the process discussed in this study differs slightly from the process which is commonly referred to as singlet fission and after which both partners are in their excited state. This also relates to the fact that the authors do not observe the formation of a coupled triplet pair state.

To me it seems that oxygen is a very special case due to its triplet ground state and I would be interested in the authors' view on this.

2) The first step which is discussed by the authors involves energy transfer from the excited singlet state of the acene to the triplet state of oxygen. As stated in references 16 and 17 provided by the authors, this is a possible process. Still, it is very unusual and some details about the mechanism of this energy transfer would be helpful for the reader.

3) The authors state on page 4 that "homogeneous SF and energy transfer to $3O_2$ occurs simultaneously". However, I couldn't find the respective time scales expected for these two processes.

4) On page 5 the authors attempt to disentangle the contribution of homogeneous and energy transfer by discussing the bimolecular nature of homogeneous SF. Heterogeneous SF is bimolecular as well but does not seem to be included in Eq. 4. Can the authors clarify the justification of Eq. 4? Furthermore, the argument how energy transfer can be determined by singlet-triplet annihilation is unclear to me and additional details would be helpful here.

5) On page 5 "the homogeneous SF rate constant is obtained as the slope of the total singlet decay rate". This seems to neglect the radiative decay of the excited singlet state. Could the authors comment on this?

6) The reasoning behind the energy gain of 0.19eV on page 5 is unclear to me.

7) I assume that the first sharp intensity drop for the lowest concentrations in Fig. 4 arises from the instrument response and the laser pulse, but it is not discussed in the manuscript.

8) Have the authors attempted to measure the effect of external magnetic fields on the time scales? This would probably be interesting as it should affect the rate of energy transfer and heterogeneous SF.

9) How can the authors exclude photooxidation of the acenes, which I would expect to occur when using high fluences in oxygen-rich environments?

Some minor comments:

a) The second sentence ("Architectures...use the solar spectrum more efficiently...and to...") and the second-last sentence ("Therefore, in favourable cases, SF outcompetes...") on page 2 seem to be incomplete.

b) The first full sentence on page 3 is unclear to me. What is the meaning of "repopulation of the solar cell sensitizer in the ground electronic state of typical SF chromophores"?

c) What is a "pseudo-first" order?

d) The ground-state bleach is not necessarily caused by the "ground-state hole left behind", but is due to the fact that not all molecules excited by the pump pulse have returned to their ground state when the probe pulse arrives (depending on the lifetime of the excited state and the pump-probe

delay). Thus, for these molecules the transition S₀-S₁ cannot be excited again by the probe, leading to reduced absorption at the position of the S₀-S₁ transition.

Reviewer #3 (Remarks to the Author):

The manuscript by Wollscheid et al describes a unique form of singlet fission in which triplets are sequentially formed on acene molecules with molecular oxygen as an intermediate. Forms of "heterofission" in which triplets form on distinct molecules is not new, nor is stepwise singlet fission with an intermediate, although the inclusion of the molecular oxygen species appears novel and is interesting. The authors do an excellent job of systematically proving the scenario that they put forth. Because of some controversy and confusion about this phenomenon (intramolecular SF in solution), further examination of the mechanism is warranted. I believe the data can be published as a fundamental observation and provides some new "tools" for measuring SF. I have some comments below that I hope will improve the clarity of the manuscript and help to provide appropriate context for a general audience.

(1) I find the first portion of the introduction to be somewhat disingenuous. For solar applications the first and most important criterion is efficiency of triplet pair production. The lifetime is secondary because there are numerous schemes that could harvest the triplets prior to the mentioned decay processes. But, if the triplet yield has no pathway above 100% and toward 200%, the motivation for singlet fission enhanced solar conversion is lost. Yet, the authors do not spend any space discussing why the yield is found to be low, or how it can be improved. What could be done on the molecular level (besides increasing the concentration) to improve the unfavorable kinetic situation? It is fine if the ultimate efficiency will remain low, but then motivation for a practical purpose should not be presented so prominently.

(2) Experimental demonstration for an "unsuitable" molecule (e.g., TIPS-tetracene) would be ideal. There is some debate about the actual triplet energy for TIPS-tetracene, and the value of 1.40 eV that is quoted seems like an unreasonably large number. How was it determined?

(3) Although the molecules presented are relatively stable, the presence of singlet oxygen may eventually cause the degradation of the entire system – can data on longer term stability be shown?

(4) The only advantage that I can imagine for actual light harvesting (besides operation in ambient environment, which is rarely necessary) might be the increased length over which triplets diffuse in this mechanism vs. when bound in the triplet pair. Could the authors estimate the effective diffusion length increase, taking advantage of the relatively large diffusivity of the intermediate O₂ in solvent?

Letter of response: Addressing the Nature Communications reviewer feedback and highlighting changes

Dear Referees,

Please find attached a revised version of our manuscript "Oxygen-Catalysed Sequential Singlet Fission" by Wollscheid et al, resubmitted for publication in Nature Communications. We would like to thank for the very positive feedback and the suggestions to improve the manuscript and readability. We have accepted and addressed all changes suggested by the reviewers.

We would like to inform that during the review process Victor Brosius (PhD student in the group of Prof. Bunz, already co-author) have synthesized TIPS-Tetracene as suggested by referee #4. Therefore, we have decided to include Victor Brosius as co-author. All authors have agreed with this change.

In the following we address the remarks of each reviewer individually.

(yellow highlighted sections in the main text indicate where changes were made)

Responses to Reviewer 1

"However, a thorough rewrite of the manuscript is required first. The present text is about twice longer than it needs to be and strikes me as confusing, repetitive, and generally disorganized. The English style will require attention, too. For instance, about half of the hyphens should disappear, along with phrases such as "Therefore, in favourable cases, SF outcompetes singlet state deactivation by fluorescence and internal conversion being intersystem-crossing of no concern for such a big $E(S) - E(T)$ energy bias ..."

Author Reply:

As suggested by the reviewer, the text has been corrected by the Springer Nature Author Services (a commercial English editing service; see please Certificate). The abstract was also shortened to below 150 words and the manuscript was formatted following Nature Communications guidelines. We were not able to significantly reduce the length of the manuscript due to the several requests and suggestions by the other referees, but the word count is well below the max number allowed by Nature Communications.

Responses to Reviewer 2

1. *"Commonly, the term singlet fission is used for the photophysical process in which two molecules are involved and interact directly (one in the ground state, the other in the excited state). Initially, both molecules are in their singlet state and in the (excited) triplet state after completion of the singlet fission process. Here, the two acenes do not directly interact, which justifies the term "oxygen-catalysed singlet fission". However, I am wondering, if the second step can still be considered as a heterogeneous singlet fission process, since in this case the oxygen, first, is not photoexcited and, second, returns to its (triplet) ground state after interaction with the ground state acene. In my view, the process discussed in this study differs slightly from the process which is commonly referred to as singlet fission and after which both partners are in their excited state. This also relates to the fact that the authors do not observe*

the formation of a coupled triplet pair state. To me it seems that oxygen is a very special case due to its triplet ground state and I would be interested in the authors' view on this."

Author Reply: Singlet fission is not necessarily related to *photoexcitation*, but describes the interaction between an excited-state with a ground-state. This is exactly what we have observed: One excited singlet ($^1\text{O}_2$) and one ground state singlet (chromophore) yield two triplet states. The process is heterogeneous because it involves molecules of different kinds. The term was already coined by Swenberg, Tedder, Webber et al in the 70s in the context of triplet kinetics between different molecules (for example, "Homogeneous and Heterogeneous Triplet-Triplet Annihilation in Anthracene-Doped Phenanthrene Crystals" by Tedder and Webber CPL 31(1975) 611-616).

Moreover, the terms "homofission" and "heterofission" have been already used in the definitive review about singlet fission by Smith and Michl (Chemical Review 110 (2010) 6891): "... *The two chromophores can be of the same kind ("homofission") or of different kinds ("heterofission")....*"

Thus, we stand by the explanation already given in the manuscript on page 4 (" $^1\text{O}_2$ interacts with a chromophore in the ground state, resulting in the T_1 chromophore and the $^3\text{O}_2$ ground-state of oxygen").

2. *"The first step which is discussed by the authors involves energy transfer from the excited singlet state of the acene to the triplet state of oxygen. As stated in references 16 and 17 provided by the authors, this is a possible process. Still, it is very unusual and some details about the mechanism of this energy transfer would be helpful for the reader."*

Author Reply: We have rewritten the paragraph on page 3/4 to include more details about the process in question:

1) In an initial energy transfer stage, the chromophore is excited to the first-excited singlet state S_1 by an incoming photon. Subsequently, it crosses to its T_1 state in a spin-allowed singlet-triplet annihilation involving the excited chromophore and $^3\text{O}_2$. The latter is thereby excited to its lowest excited-singlet state ($^1\text{O}_2$). This process is a well-established method used in the photosensitised production of $^1\text{O}_2$ and is exothermic whenever the singlet-triplet energy gap of the chromophore exceeds the singlet-triplet energy gap of O_2 (0.977 eV):¹⁶

3. *"The authors state on page 4 that "homogeneous SF and energy transfer to $^3\text{O}_2$ occurs simultaneously". However, I couldn't find the respective time scales expected for these two processes."*

Author Reply: An explanation regarding the time scales for the individual processes has been added to the SI (page 2):

The reaction rates k for homogeneous SF and energy transfer to $^3\text{O}_2$ depend on the concentration of the reactants, which are TIPS-Pn and $^3\text{O}_2$, respectively. This is shown in the main text (Equation 4). Thus, the timescales can be calculated as $\tau = 1/k$ for any given concentration. For homogeneous SF, the examined concentration range of TIPS-Pn is 0.02 to 160 mM. Using the reported literature value of $k_{\text{SF}} = 2.18 \times 10^9 \text{ (M s)}^{-1}$ (SI Table 3), this corresponds to a timescale of 2.30 μs to 2.87 ns. Regarding the energy transfer step, the reported literature rate of $k_1 = 3.12 \times 10^{10} \text{ (Ms)}^{-1}$ for atmospheric conditions ($[\text{O}_2] = 1.81 \text{ mM}$) corresponds to a time constant of $\tau_1 = 17.7 \text{ ns}$. In deaerated solutions, energy transfer does not occur at all.

A new sentence was also added at page 5 in the main text:

The individual timescales for these two sub-reactions can be vastly different, depending on the respective concentrations of the reaction partners (see SI).

4. *“On page 5 the authors attempt to disentangle the contribution of homogeneous and energy transfer by discussing the bimolecular nature of homogeneous SF. Heterogeneous SF is bimolecular as well but does not seem to be included in Eq. 4. Can the authors clarify the justification of Eq. 4? Furthermore, the argument how energy transfer can be determined by singlet-triplet annihilation is unclear to me and additional details would be helpful here.”*

Author Reply: In Eq. 4 the total singlet decay rate k_{tot} is discussed, meaning all possible decay channels occurring out of the initially formed S_1 state. The heterogeneous SF step does not include the S_1 state of the chromophore (see please Figure 2a, the blue arrow). Therefore, the term singlet-triplet-annihilation refers to the mechanism of the energy transfer process, during which the S_1 state of TIPS-Pn and 3O_2 annihilate. The term “singlet-triplet annihilation” has been used in textbooks like Swenberg, C. E.; Geacintov, N. E. in Birks, J. B. Ed.: Organic Molecular Photophysics, Wiley 1973, Chapter 10, pages 489-564.

We have rewritten this part at page 5:

It is possible to disentangle under excess concentrations the individual values by the bimolecular nature of homogeneous SF (Figure 2b, red arrow) and energy transfer between the chromophore (S_1) and the molecular oxygen (3O_2) (Figure 2a, green arrow).

We have also reorganized and streamlined the text around equation 4 to clarify its justification and meaning.

5. *“On page 5 “the homogeneous SF rate constant is obtained as the slope of the total singlet decay rate”. This seems to neglect the radiative decay of the excited singlet state. Could the authors comment on this?”*

Author Reply: The radiative decay rate is a constant in equation 4, therefore, when the total singlet decay rate is plotted against concentration of the oxygen (or the chromophore), it will be the intercept of that plot. In order to make that point more clear, we have expanded and reorganized this part at page 5:

..., which are experimentally known. In contrast, the unimolecular singlet decay rate constant (k_R) does not depend on the concentrations of the ground-state chromophore or molecular oxygen and explains the intercept of k_{tot} as function of the concentrations of both reaction partners (chromophore S_0 and triplet oxygen).

6. *“The reasoning behind the energy gain of 0.19eV on page 5 is unclear to me.”*

Author Reply: The reactants of the energy transfer step are the excited TIPS-Pn Singlet and molecular oxygen in its ground state. Therefore, their total energy relative to their respective ground states is 2.02 eV. The products of the reaction, T_1 -TIPS-Pn and 1O_2 , are combined 1.83 eV above their respective ground states. Thus, an energy difference of 0.19 eV is observed (Compare also Fig. 2a). To make the calculation more clear, the energy of the T_1 state of TIPS-Pn was added on page 5:

The first excited singlet state of TIPS-Pn locates at 2.02 eV and the first triplet state 0.85 eV above its ground-state (Figure 2)

7. *"I assume that the first sharp intensity drop for the lowest concentrations in Fig. 4 arises from the instrument response and the laser pulse, but it is not discussed in the manuscript."*

Author Reply: The intensity drop arises from the decay of the initial S_1 state. In order to clarify this, the paragraph on page 10 was rephrased:

Here, the dynamics depend on the S_0 concentration. Within the range of $0.1 \text{ mM} \leq [S_0] \leq 6 \text{ mM}$, the TA signal evolves three-exponentially in time. A sharp initial intensity drop within $<0.1 \mu\text{s}$ is observed, corresponding to the decay of the S_1 state. Subsequently, a rising component in the T_1 signal as well as a local μs maximum can be identified. Increasing the chromophore concentration accelerates this component and thus shifts this local μs maximum to earlier probe delays.

Furthermore, the Methods section on page 15 was expanded to include the value of the instrument response function:

Transient absorption was measured in a commercial setup (EOS from Ultrafast Systems). Pump beams were derived from a Ti:Sa amplified femtosecond laser system (Coherent Astrella, 4 kHz, 90 fs pulse duration, 1.5 mJ/pulse) and probed using a triggerable on-demand supercontinuum laser. The instrument response function reaches a FWHM of 350 ps.

8. *"Have the authors attempted to measure the effect of external magnetic fields on the time scales? This would probably be interesting as it should affect the rate of energy transfer and heterogeneous SF."*

Author Reply: We agree that investigating the effects of magnetic fields on dynamics of these photoreactions would further demonstrate the processes contributing to the mechanism. Unfortunately, we cannot carry out this experiment on a reasonable short time and we believe that the nature of the different stages is sufficiently clear, in particular after the additional measurements we have done on TIPS-Tetracene, as requested by referee#4. We also argue that the influence of external magnetic fields has a more straightforward interpretation in oriented crystals, where however heterogeneous SF would not occur due to hindered diffusion.

9. *"How can the authors exclude photooxydation of the acenes, which I would expect to occur when using high fluences in oxygen-rich environments?"*

Author Reply: We did not observe any photooxydation for the samples investigated in this work during the measurement times. Stationary absorption spectra were measured before and after all transient measurements and no change in the shape or intensity is observed. A corresponding explanation as well as an additional figure were added to the SI at page 5.

The stationary spectra of TIPS-Pn and TDCI₄ before and after transient absorption measurements show a perfect overlap, thus displaying great photostability (Figure SI 2a and b). In contrast to this, unsubstituted pentacene degrades almost completely when excited for 2h under measurement conditions (Figure SI 2c). Measurements were carried out with a mean energy per pulse of about 400 nJ at a 2 kHz repetition rate. The spot diameter was 0.3 mm. TIPS-Pn, TDCI₄ and pentacene were excited at 680, 620 and 575 nm, respectively.

Figure SI 2: Absorption Spectra of a) TIPS-Pentacene in THF, b) TDCl₄ in toluene and c) Pentacene in THF before and after irradiation for 2h under ambient measurement conditions.

a) *"The second sentence ("Architectures...use the solar spectrum more efficiently...and to...") and the second-last sentence ("Therefore, in favourable cases, SF outcompetes...") on page 2 seem to be incomplete."*

Author Reply: The respective sentences have been corrected:

Architectures based on organic materials and amorphous semiconductors are interesting due to an increased charge/photon ratio, efficient use of the solar spectrum and low fabrication costs.

Therefore, in favourable cases, SF outcompetes singlet-state deactivation by fluorescence and internal conversion, with intersystem crossing being of no concern for such a big $E(S_1) - E(T_1)$ energy bias:

b) *"The first full sentence on page 3 is unclear to me. What is the meaning of "repopulation of the solar cell sensitizer in the ground electronic state of typical SF chromophores"?"*

Author Reply: Following the suggestions of referee #1 and #4 to shorten and improve that part, we have removed that sentence.

c) *"What is a "pseudo-first" order?"*

Author Reply: A pseudo-first reaction order is used to describe a bimolecular reaction, in which one of the reactants is in excess. Therefore, the concentration of the latter can be treated as a constant, which significantly simplifies analytical solutions of rate equations. In the manuscript, this approach is used for reactions of excited species with other reactants. This approach is justified, as only a small fraction of the chromophores is excited with the pump pulse. The paragraph on page 4 has been rephrased to explain this behaviour:

After excitation, homogeneous SF and energy transfer to $^3\text{O}_2$ (with bimolecular rate constants k_1 and k_{SF} , respectively) occur simultaneously. Homogeneous SF and energy transfer occur out of the same S_1 species. For both reactions, the respective reaction partner for the S_1 species (S_0 and $^3\text{O}_2$) is in excess. Thus, their concentration can be assumed remain constant over time, resulting in a pseudo-first order reaction for both processes.

d) *“The ground-state bleach is not necessarily caused by the “ground-state hole left behind”, but is due to the fact that not all molecules excited by the pump pulse have returned to their ground state when the probe pulse arrives (depending on the lifetime of the excited state and the pump-probe delay). Thus, for these molecules the transition S_0 - S_1 cannot be excited again by the probe, leading to reduced absorption at the position of the S_0 - S_1 transition.”*

Author Reply: The referee is of course right and that was the way the phrase was meant to be understood. Thus, we have rephrased the corresponding section:

The bleach arises from molecules being promoted to excited states, which consequently depopulates the ground state. As TA monitors absorption differences, a “decrease” in the signal is observed, with its spectral contributions being equal to the UV-Vis absorption spectrum of the chromophore with a negative sign.

Responses to Reviewer 3

1. *"I find the first portion of the introduction to be somewhat disingenuous. For solar applications the first and most important criterion is efficiency of triplet pair production. The lifetime is secondary because there are numerous schemes that could harvest the triplets prior to the mentioned decay processes. But, if the triplet yield has no pathway above 100% and toward 200%, the motivation for singlet fission enhanced solar conversion is lost. Yet, the authors do not spend any space discussing why the yield is found to be low, or how it can be improved. What could be done on the molecular level (besides increasing the concentration) to improve the unfavorable kinetic situation? It is fine if the ultimate efficiency will remain low, but then motivation for a practical purpose should not be presented so prominently."*

Author Reply: We have rewritten the motivation ("practical purpose" was removed) as suggested and a new paragraph has been added to page 12 to discuss the yields:

The upper panel in Figure 5 shows the quantum yield for energy transfer and homogeneous SF. Both processes ultimately lead to the formation of two triplets (see above). The upper panel reveals that homogeneous SF can reach quantum yields of 100% only at high acene concentrations on the order of 0.1-1 M. In contrast, sequential SF occurs even at very low concentrations, albeit with lower quantum yield. By increasing the concentration of $^3\text{O}_2$ from 1.81 mM (ambient conditions²⁵) to 5.3 mM in a 1.5 mM TIPS-Pn solution, the quantum yield rises from 16% to 46% (see SI). Apart from this strategy, the quantum yield could also be improved by finding other suitable catalysts and/or introducing chemical modifications to the SF chromophore, which favour covalent or non-covalent bonding between acene and catalyst.

Additionally, Section 6.1 of the SI has been modified to include the calculation of the quantum yield:

This is directly linked to the increase of the $^3\text{O}_2$ concentration, as k_{SF} and k_{R} must remain constant. Thus, the difference between k_{tot} for O_2 -enriched and ambient conditions yields (see Eq. 4)

$$\Delta k_{\text{tot}} = k_1([\text{}^3\text{O}_2](\text{enriched}) - [\text{}^3\text{O}_2](\text{ambient})) \quad (4)$$

Using the known values for k_1 as well as the known oxygen concentration for atmospheric conditions (SI Table 3), the concentration for the O_2 -enriched sample can be calculated as 5.0 mM. Subsequently, the quantum yield for the energy transfer process can be calculated as

$$QY = \frac{k_1[\text{}^3\text{O}_2]}{k_{\text{tot}}} \quad (5)$$

Based on that, the quantum yield doubles from 21.6 % to 43.4 % when increasing the $^3\text{O}_2$ concentration from 1.81 to 5.0 mM.

2. *"Experimental demonstration for an "unsuitable" molecule (e.g., TIPS-tetracene) would be ideal. There is some debate about the actual triplet energy for TIPS-tetracene, and the value of 1.40 eV that is quoted seems like an unreasonably large number. How was it determined?"*

Author Reply: We thank the reviewer for this comment. We have synthesized TIPS-Tetracene and measured a solution with 0.8 mM in THF under the same conditions as TIPS-Pn. As we have predicted in Fig.6, the initial step in the sequential SF i.e. the energy transfer takes place with very low yield, but the second step i.e. the heterogeneous SF does not. These results were added to the SI:

6.2 Effects of oxygen on the singlet decay of TIPS-tetracene

The triplet energy of TIPS-tetracene (Figure 6) does not allow for sequential SF. This is confirmed by TA measurements of 0.8 mM TIPS-tetracene in deaerated and oxygen-equilibrated THF. In both cases, an initially formed singlet ESA with peaks at 415 and 470 nm is observed (Figure SI a and b). Within 40 ns, a triplet ESA centred at 510 nm is formed. Under ambient conditions, this signal decays completely whereas it remains constant in a deaerated solution. A global fit shows a slight acceleration of a monoexponential singlet decay from 13.3 ± 0.1 ns (deaerated) to 10.8 ± 0.1 ns (ambient), as seen for the transients at 420 nm (Figure SI c). This corresponds to an acceleration of the reaction rate of $1.72 \cdot 10^7$ s⁻¹. The triplet signal decays with a time constant of 169 ± 6 ns under ambient conditions. In a deaerated solution, no decay can be observed within 4 μ s (Figure SI 7d), comparable to TIPS-Pn (Figure 3b).

The acceleration of the singlet decay for ambient conditions is in accordance with the diffusion rate of molecular oxygen (SI Table 3). This suggests that the same energy transfer process observed in TIPS-Pn and TDCI₄ takes place (SI Tables 3 and 4). However, in contrast to the results presented in the main text (Figure 3b), TIPS-tetracene shows no rising component in the triplet signal intensity. Thus, it can be concluded that no heterogeneous SF occurs. Both findings are in agreement with the presented model for sequential SF as well as calculated energy levels (Figure 6).

Figure SI 7: Transient spectra at selected delays for TIPS-tetracene in THF a) under ambient conditions and b) in a deaerated solvent. c) Transients at 420 nm for 0.8 mM TIPS-tetracene in THF under ambient conditions (red) and in a deaerated solvent (blue) with respective fits. d) Transients at 510 nm for 0.8 mM TIPS-tetracene in THF under ambient conditions (red) and in a deaerated solvent (blue) with respective fits.

Moreover, the missing reference for TIPS-Tn was added (ref. 14 of the manuscript). The value was calculated using TD-DFT within the Tamm-Dancoff-Approximation with the M06-2X/6-31G* basis set. A new sentence was also added at page 14 in the main text:

This condition is not fulfilled for TIPS-Tn, which does not show triplet formation by heterogeneous SF (see SI).

The caption of Figure 6 was also expanded:

According to this model, catalysed SF should not occur in TIPS-Tn, as the T_1 state lays higher than 1O_2 in energy (see SI for measurements done with and without O_2).

Finally, the 1H NMR of the synthesized TIPS-Tn was added to the SI

Figure SI 17: TIPS-Tetracene: 1H NMR (500 MHz, DICHLOROMETHANE- d_2) δ 9.33 (s, 2H), 8.67 – 8.62 (m, 2H), 8.07 – 8.03 (m, 2H), 7.60 – 7.55 (m, 2H), 7.51 – 7.48 (m, 2H), 1.41 – 1.26 (m, 42H)

3. “Although the molecules presented are relatively stable, the presence of singlet oxygen may eventually cause the degradation of the entire system – can data on longer term stability be shown?”

Author Reply: The absorption spectra before and after the measurements do not change. Please refer to our reply to question 9 of reviewer 3.

4. “The only advantage that I can imagine for actual light harvesting (besides operation in ambient environment, which is rarely necessary) might be the increased length over which triplets diffuse in this mechanism vs. when bound in the triplet pair. Could the authors estimate the effective diffusion length increase, taking advantage of the relatively large diffusivity of the intermediate O_2 in solvent?”

Author Reply: An additional section on diffusion length was added to the SI. The diffusion length increases by a factor of 68.3 for oxygen compared to TIPS-Pn.

Diffusion Length

The diffusion length L can be calculated as $L = \sqrt{D\tau}$ with the diffusivity D and the lifetime τ . D can be approximated using the Stokes-Einstein-Relation $D = \frac{k_B T}{6\pi\eta r}$ with $r(\text{O}_2) = 2 \text{ \AA}$, $r(\text{TIPS-Pn}) = 4.13 \text{ \AA}$ and $\eta = 0.48 \times 10^{-3} \text{ kg/(ms)}$. The lifetimes τ were taken from ref. 22 for oxygen or calculated as $1/k_R$ (SI Table 3), respectively. Thus, a Diffusion length of 246 nm for oxygen and 3.6 nm for TIPS-Pn is obtained. This shows that the diffusion length increases by a factor of 68.3 for oxygen compared to TIPS-Pn.

A new sentence was also added at page 4 in the main text:

Additionally, the energy transfer to the long-lived $^1\text{O}_2$ species (decay time of 26.7 μs in benzene) greatly increases the diffusion length of the excited singlet species from 3.6 nm to 246 nm (see SI).

Best regards,

Tiago Buckup

REVIEWERS' COMMENTS:

Reviewer #1 (Remarks to the Author):

The writing is considerably improved, but the paper is still much wordier and longer than it needs to be. This apparently is what the authors prefer, even though it hurts their case with the readers, and I would not object to publication after one additional careful reading and removal of small errors and infelicities. I did not try to note them all, that is the authors' job. For illustration: in the first sentence of the abstract, I would say "of a singlet exciton into two triplet excitons" instead of "of an excited singlet exciton into two triplet states" - aren't all singlet excitons excited? In line 51, an "l" is missing in the name of the compound. In line 88, "lays" is wrong, it should be "lies". In Figure 3, I doubt that "catalysation" is an English word. Should it not be "catalysis"?

As I wrote earlier, the science is good and interesting, and I do not need to see the manuscript again.

Reviewer #2 (Remarks to the Author):

The changes made in the text, the additional experiments and the more detailed discussion of the analysis have significantly improved the manuscript and I recommend publication.

Reviewer #3 (Remarks to the Author):

The revision has added some negative controls (TIPS-tetracene) and information that puts the results in better context. I am satisfied with the changes made and recommend no additional modifications.

Letter of response: Addressing the Nature Communications reviewer feedback and highlighting changes

Dear Reviewer,

Please find attached a revised version of our manuscript "Oxygen-Catalysed Sequential Singlet Fission" by Wollscheid et al, resubmitted for publication in Nature Communications. We thank the reviewers for the very positive feedback. We have accepted and addressed all changes suggested by the reviewer.

In the following we address the remarks individually.

(yellow highlighted sections in the main text indicate where changes were made)

Responses to Reviewer 1

"The writing is considerably improved, but the paper is still much wordier and longer than it needs to be. This apparently is what the authors prefer, even though it hurts their case with the readers, and I would not object to publication after one additional careful reading and removal of small errors and infelicities. I did not try to note them all, that is the authors' job. For illustration: in the first sentence of the abstract, I would say "of a singlet exciton into two triplet excitons" instead of "of an excited singlet exciton into two triplet states" - aren't all singlet excitons excited? In line 51, an "l" is missing in the name of the compound. In line 88, "lays" is wrong, it should be "lies". In Figure 3, I doubt that "catalysation" is an English word. Should it not be "catalysis"?"

Author Reply: As suggested by the reviewer, minor errors have been corrected. The text in figure 3 has been changed to "Catalytic Effect".

Best regards,

Tiago Backup